# Mathematical Optimisation of Magnetic Nanoparticle Diffusion in the Brain White Matter

**DOI:** 10.3390/ijms24032534

**Published:** 2023-01-28

**Authors:** Tian Yuan, Yi Yang, Wenbo Zhan, Daniele Dini

**Affiliations:** 1Department of Mechanical Engineering, Imperial College London, London SW7 2AZ, UK; 2School of Engineering, King’s College, University of Aberdeen, Aberdeen AB24 3UE, UK

**Keywords:** brain tissue, diffusion, drug delivery, magnetic nanoparticle, mathematical modelling

## Abstract

Magnetic nanoparticles (MNPs) are a promising drug delivery system to treat brain diseases, as the particle transport trajectory can be manipulated by an external magnetic field. However, due to the complex microstructure of brain tissues, particularly the arrangement of nerve fibres in the white matter (WM), how to achieve desired drug distribution patterns, e.g., uniform distribution, is largely unknown. In this study, by adopting a mathematical model capable of capturing the diffusion trajectories of MNPs, we conducted a pilot study to investigate the effects of key parameters in the MNP delivery on the particle diffusion behaviours in the brain WM microstructures. The results show that (i) a uniform distribution of MNPs can be achieved in anisotropic tissues by adjusting the particle size and magnetic field; (ii) particle size plays a key role in determining MNPs’ diffusion behaviours. The magnitude of MNP equivalent diffusivity is reversely correlated to the particle size. The MNPs with a dimension greater than 90 nm cannot reach a uniform distribution in the brain WM even in an external magnitude field; (iii) axon tortuosity may lead to transversely anisotropic MNP transport in the brain WM; however, this effect can be mitigated by applying an external magnetic field perpendicular to the local axon track. This study not only advances understanding to answer the question of how to optimise MNP delivery, but also demonstrates the potential of mathematical modelling to help achieve desired drug distributions in biological tissues with a complex microstructure.

## 1. Introduction

Degenerative nerve diseases, including Alzheimer’s disease and brain cancers, are increasingly threatening human health around the world, particularly for the population over 60 years old [1]. Brain diseases have proven difficult to treat with conventional drug delivery procedures. The disappointing effectiveness is largely due to (i) the limitations of the impermeable nature of the blood–brain barrier (BBB) and (ii) the compact microstructure of nerve fibres in the brain white matter (WM) [2]. Encapsulating drugs inside and serving as vehicles, nanoparticles (NPs) offer great flexibility to optimise size and surface properties that enable drugs to cross the BBB [3,4]. To date, numerous studies have been conducted to improve the NP fabrication and fine-tune NP properties for enhancing this transvascular transport [5,6,7,8,9].

However, the latter limitation remains. NPs need to deliver sufficient drugs to the lesion to ensure adequate drug exposure for effective treatment. Their transport in the brain highly depends on the tissue microstructure. The cable-like nerve fibres, as shown in Figure 1, would guide the NPs to undesired directions and locations, resulting in uncontrollable drug distribution and low drug concentration in the target location [10,11]. This drawback becomes more serious for drug delivery to those lesions embedded deep in the brain tissue.

Magnetic nanoparticles (MNPs) that contain a paramagnetic core (e.g., iron oxide) present great potential to overcome the latter limitation, since their motion can be steered by an externally applied magnetic field [13,14]. Their feasibility in the treatments of brain circulation system diseases and brain tumours has been explored in some preclinical studies by means of mathematical modelling and experimental observations. For example, Rotariu et al. developed a mathematical model to investigate different techniques to focus small MNPs within the microvasculature of tumours [15]. Sharma et al. mathematically captured the transport behaviours of a cluster of MNPs in a blood vessel to study the application of magnetic drug targeting (MDT) [16]. Kenjeres et al. investigated the concept of the targeted delivery of magnetic pharmaceutical drug aerosols in the human upper and central respiratory system, also by mathematical modelling [17]. The fundamental theory of these mathematical models is to calculate the trajectories of MNPs in different environments based on the specific acting forces that determine the NPs’ movements, e.g., magnetic force, buoyancy force, and Newton’s second law. Regarding experimental studies, except for extensive investigations on coating Fe_3_O_4_ NPs with, e.g., poly(ethylene glycol) (PEG) and carboxymethyl cellulose (CMC), to increase their capability of BBB penetration, researchers also found that magnetically labelled cancer cells can be killed by magnetic iron oxide NPs when subjected to oscillating gradients in a strong external magnetic field [18]. More investigations on MNP applications are reported in Ref. [19].

All these studies have shown the important role of magnetic field intensity and topography in manipulating the transport behaviours of MNPs in tissues, blood vessels, and the BBB. However, the transport of MNPs in the brain parenchyma is still unclear. The lack of this knowledge would, on the one hand, lead to the misjudgement of treatment protocols, and on the other hand, limit the development of MNPs for clinical use, particularly for treatments against degenerative nerve diseases [20], which mainly occur in nerve-fibre-rich WM.

The present study aims to tackle this challenge by investigating the diffusion phenomenon of MNPs in the brain WM. Given that mass transport in the brain interstitium is governed by diffusion rather than bulk movement with the interstitial fluid flow [21], the transport efficiency of the MNPs can be explicitly represented by a diffusion coefficient tensor (**D**) [22], which can be statistically calculated by monitoring the MNP trajectories [23]. Due to the lack of valid experimental means with ultra-high resolution and a frame rate to directly track the MNPs’ movements in deep brain tissue, we adapted a mathematical framework capable of capturing the diffusion process of NPs in the brain WM to reproduce the diffusion process of MNPs in an idealised 3D model of brain WM microstructure by further considering the factor of magnetic force. A group of systematic parametric studies were conducted using this model to examine how the key factors of this drug delivery method affect the equivalent diffusion coefficient tensor of MNPs in the brain WM; these include particle size, the particle’s magnetic property, magnetic field intensity, magnetic field gradient, magnetic field direction, and the tissue microstructure. The results provide feasible strategies to optimise the magnetic NP-mediated drug delivery to the brain tissues.

## 2. Materials and Methods

### 2.1. Mathematical Model

Established based on Newton’s second law, the mathematical model consists of a set of governing equations for the major forces acting on the NPs, including thermal motion, particle–particle interaction, particle–fluid interaction, and particle–axon interaction. Its capability and accuracy of predicting the diffusion coefficient of NPs in the brain WM have been validated and reported in our previous study [24]. In this study, the model was further developed to consider the magnetic force. The mathematical model in this study thus includes the Brownian force (thermal motion, Equation (Equation 1)), drag force (resistance due to the fluid viscosity, Equation (Equation 2)), and magnetic force (Equation (Equation 3)).
(1)$$\begin{array}{cc} \; & F_{B}=\Phi \sqrt{\frac{12\pi K_{B}\mu Tr_{p}}{\delta t}} \end{array}$$
(2)$$\begin{array}{cc} \; & F_{D}=6\pi \mu r_{p}\left(v_{\mathrm{flow}\;}-v_{\mathrm{particle}\;}\right) \end{array}$$
(3)$$\begin{array}{cc} \; & F_{M}=\frac{V\Delta \chi }{\mu_{0}}\left(B\cdot \nabla \right)B \end{array}$$
where $F_{B}$, $F_{D}$, and $F_{M}$ are the Brownian force, drag force, and magnetic force, respectively; $k_{B}$ is the Boltzmann constant; μ is the dynamic viscosity of the fluid; *T* is the absolute temperature of the fluid; δt is the time step used to calculate the Brownian force; Φ is a Gaussian random number with zero mean and unit variance to take the randomness of Brownian motion into account; $r_{p}$ is the radius of the particle; $v_{\mathrm{flow}}$ is the velocity of fluid flow; $v_{\mathrm{particle}}$ is the velocity of the particle; *V* is the particle volume, Δχ is the difference in magnetic susceptibilities between the particle and the surrounding medium; $\mu_{0}=4\pi \times 10^{-7}H/m$ is the permeability of the vacuum; and B is the applied magnetic field.

Since this study is focused on the diffusion phenomenon only, the fluid was assumed to be static, i.e., $v_{\mathrm{flow}\;}=0$. Obeying Newton’s second law, the displacement of a particle *i* is
(4)$$dr_{i}=\left[\sum_{i=1}^{N}\left(F_{Bi}+F_{Di}+F_{Mi}\right)\right]{\left(\Delta t\right)}^{2}$$
where $dr_{i}={\left(dx_{i},dy_{i},dz_{i}\right)}^{T}$ is the displacement vector of the *i* th particle, and Δt is the time step. The equivalent diffusion coefficient in the X direction can be then obtained by
(5)$$\begin{array}{cc} D_{xx} & =\langle R_{xx}^{2}\rangle /2t \end{array}$$
(6)$$\begin{array}{cc} \langle R_{xx}^{2}\rangle & =\sum_{i=1}^{n}{\left(dxx_{i}\right)}^{2} \end{array}$$
where $\langle R_{xx}^{2}\rangle$ is the average mean square displacement (MSD) of all the particles in the X direction; dxx is the displacement of an NP in the X direction; *n* is the number of NPs in the system; and *t* is the diffusion time.

The equivalent diffusion coefficients in the Y and Z directions have the same definitions as in Equations (Equation 5) and (Equation 6), using the parameters for the Y and Z directions, respectively. Please note that the main axis of the coordinate system is placed parallel to the axon tracts in the computation domain. This enables the use of the diffusion ellipsoid, with which only the three diagonal elements of $D_{xx}$, $D_{yy}$, and $D_{zz}$ are needed.

### 2.2. Geometric Model

The 3D geometry of the brain WM microstructure was reconstructed by sweeping a representative cross-sectional geometry of brain WM [24] along the axon tracts. Figure 1 of Ref. [25] shows that axons appear “wavy” or undulated under in situ length conditions when visualised using neurofilament immunohistochemistry. We therefore reconstructed the 3D microstructure, as shown in Figure 2A (model dimension: 20 μm × 20 μm × 20 μm). Tortuosity (τ) is an important geometric parameter that describes the longitudinal shape of axons. It is defined as the ratio of axon length to the distance between the axon’s two endpoints. The statistical data of axon tortuosity presented in Ref. [26] demonstrate that most axons have tortuosity ranging from 1.0 (i.e., straight) to 1.3. The average distance between axons is 100 nm and the tissue porosity is about 0.3, both of which are located in the experimental range [27,28]. Based on this information, we reconstructed the 3D geometry. Details on generating representative cross-sectional MW geometry are reported in Ref. [24].

### 2.3. Material Properties

Although we did not consider the fluid flow, the fluid viscosity is important to the particle diffusion behaviour. We thus adopted the viscosity of measured interstitial fluid $3.5\times 10^{-3}$ Pa·s [29]. Based on the practical applications of NPs used to treat brain diseases [2,30], the NP size normally does not exceed 100 nm, because measurements show that the distance between neurons is within 38–64 nm [27]. Therefore, in this study, NPs with diameters of 10 nm, 30 nm, 50 nm, 70 nm, and 90 nm were investigated. It is worth noting that the magnetic susceptibility of MNPs is not constant, and is highly dependent on the particle size, e.g., MNPs smaller than 50 nm can be super-paramagnetic (χ≫1) [31]. In order to capture the complex effects of the numerous parameters (magnetic field intensity (*B*), the gradient of magnetic field intensity ∇B, particle size (*d*), and magnetic susceptibility (χ)) while they themselves are mutually coupled, we here introduce a new parameter: magnetic force density ($f=\chi B\nabla B/\mu_{0}$ [N/m^3^]). The advantage of this parameter is that the parametric study can be significantly simplified by reducing the number of parameters needed to capture the physical system. Furthermore, from the aspect of clinical applications, the values of χ,B,and∇B can be freely chosen only if they satisfy the value of $\mu_{0}f$ (note that $\mu_{0}$ is the permeability of the vacuum, a constant value), which also significantly simplifies the clinical protocols. The rest of the parameters include temperature (310 K, i.e., normal body temperature) and Boltzmann’s constant ($k_{B}=1.38\times 10^{-23}$ J/K [32]).

### 2.4. Boundary Conditions

The particles were assumed to undergo diffuse scattering when they hit the axons, and they would move out of the computational domain when they reached the boundaries [24]. The biochemical interactions between the axons and MNPs, e.g., endocytosis, were not considered, as this study focuses on the measurement of diffusion coefficient.

### 2.5. Simulation Setup

The hex element was adopted as a mesh in the microstructural geometry, as shown in Figure 2D. According to the mesh sensitivity test, the gaps between axons should contain at least two meshes, which was the criteria for choosing element size in this study. About 1,240,000 elements were created in the geometric models. At t=0 s, 125,000 MNPs were released from a cubic domain at the centre of the model to mimic the transportation process of drug molecules from the injection site. This number of NPs was selected after a sensitivity study, which is adequate to obtain statistically stable results for calculating the MSD, as defined in Equation (Equation 6). Note that capturing the Brownian motion needs a fine time step, which also depends on the size of particles, so time step tests are essential in different occasions before choosing the value of the time step. We also conducted time step independence tests for each model. COMSOL Multiphysics 6.0 software was used to solve the mathematical model and calculate the trajectories of the MNPs. The linear solver was set as the Multifrontal Massively Parallel Sparse direct solver (MUMPS) and the Automatic Newton method was chosen as the nonlinear solver [33].

## 3. Results

We first validated the mathematical model with experimental evidence. Then, we evaluated the MNPs’ diffusion behaviours under the baseline delivery conditions to observe how the external magnetic field controls the MNP trajectory. This was followed by the analyses of the individual effect of particle size, magnetic force density, axon shape (tortuosity), and magnetic field direction on the diffusion coefficient tensor.

### 3.1. Model Validation

Due to the absence of experimental measurement on the diffusion coefficient of MNPs in the brain microenvironment, we validated the present theoretical model by using it to model the experiments reported in Refs. [34,35,36,37]. In Refs. [34,35,36], the diffusion behaviours of 10 nm and 100 nm MNPs in blood were monitored, and the corresponding Ds of the MNPs were calculated. Comparison of these data will validate the precision of the mathematical model. In Ref. [37], the *D* of gadobutrol (cerebrospinal fluid (CSF) tracer with a hydraulic diameter ($d_{H}$) of 2 nm) in the brain WM was calculated based on MRI analysis and partial differential constrained optimisation. Although non-magnetic NPs were applied, this experiment was able to validate the precision of the mathematical model excluding the magnetic force term and geometric model in the present study. Therefore, we chose these experimental data to conduct the model validation study. The experimental parameters and their respective Ds are summarised in Table 1.

By calculating the trajectories of the individual NPs using the mathematical model, we obtained the time–MSD curves of the NPs in each experiment (see Figure 3). According to Equation (Equation 5), the slopes of the curves’ stable phases were used to calculate the theoretical equivalent diffusion coefficient [24], as presented by $D_{Sim}$ in Table 1. The comparisons in Table 1 show that the theoretical results agree well with the experimental results. It is worth mentioning that the Ds obtained in the experiments are smaller than the Ds in the simulations. This could be attributed to the simplification of the mathematical model, as the resistance to particle movement from other factors, e.g., proteins and fibres, in the real microenvironment cannot be explicitly described. However, this limitation can be partly mitigated by adopting the viscosity of the measured interstitial fluid, as mentioned in Section 2.3.

### 3.2. Diffusion Behaviours of MNPs in the Brain WM

We monitored the diffusion behaviours of 50 nm MNPs under three conditions, namely, f=0 (no external magnetic field), $f=2000/\mu_{0}$, and $f=4000/\mu_{0}$, respectively, which are located in the practical ranges, since the maximum values of practically applied *B*, ∇B, and χ could reach 10 T, 100 T/m, and 200,000, respectively [38,39,40]; the values of 2000 and 4000 are within the application range. To focus on the impact of the external magnetic field, the geometry with τ=1 was adopted. Figure 3(A1–C1) show the MSD of the MNPs under these three conditions as a function of time, respectively. To more clearly interpret the mechanism behind the curves, Figure 3(A2–C2,A3–C3) present the final distributions of the MNPs (*t* = 0.08 s) from the top view and side view of the computational domain.

A rapid increase in MSD can be found in all directions at the beginning of diffusion owing to the dense MNPs and violent collisions between the MNPs. With the MNPs dispersing into the domain, interactions determining the particle motion would gradually reach dynamic equilibrium. This leads to a stable diffusion phase which is reflected as a linear increase in MSD with time, as shown in Figure 3(A1–C1), where the slop can be used to calculate the equivalent diffusion coefficient (Equation (Equation 5)). Comparisons between the time courses of MSD demonstrate the significant impacts of external magnetic field on the D of MNPs. $D_{zz}$ is much greater than $D_{xx}\;\mathrm{and}\;D_{yy}$ when no external magnetic field is applied. This is because MNPs in the Z direction would experience less resistance that is induced by the presence of axons, as shown in Figure 3(A3). $D_{xx}\;\mathrm{and}\;D_{yy}$ are almost equal, since the microstructure of the brain WM is nearly isotropic in the transverse plane, as shown in Figure 3(A2). Moreover, the external magnetic field in the XY direction can greatly increase $D_{xx}\;\mathrm{and}\;D_{yy}$. Figure 3(B1,C1) show that $D_{xx}=D_{yy}\approx D_{zz}$ when $f=2000/\mu_{0}$ and $D_{xx}=D_{yy}>D_{zz}$ when $f=4000/\mu_{0}$. Therefore, the MNP displacement in the XY direction increases with *f* (Figure 3(A2–C2)), resulting in the MNP distribution becoming more spherical, as shown in Figure 3(A3–C3).

### 3.3. Effect of Particle Size and Magnetic Force Density

Figure 4A–E show the effect of magnetic force density on the D of MNPs of different sizes. For convenience, we set the horizontal axis as $\mu_{0}f$ as it equals χB∇B, and thus, can directly determine the magnetic field and susceptibility of MNPs. We obtained the following results by comparing the results in these figures:1.When f=0 (i.e., without an external magnetic field), D decreases with the particle size due to the decreased ratio of Brownian force (Equation (Equation 1)) to drag force (Equation (Equation 2)).2.The size of MNPs plays a key role. For the MNPs with sizes ranging from 10 nm to 70 nm, the external magnetic field can override the impact of axons, and result in an isotropic D (i.e., $D_{xx}=D_{yy}=D_{zz}$). Comparisons further show that a lower magnetic force density is required for larger MNPs to achieve an isotropic D. The magnitude of isotropic diffusivity was also found to be reversely correlated to particle size.3.The diffusion anisotropy of 90 nm MNPs increases with the magnetic force density, indicating an isotropic D does not exist. This is because the MNPs with a comparable dimension as the average distance between axons (100 nm in this geometry model) are difficult to transversely diffuse in the space between axons (see the initial points in Figure 4E, which shows low $D_{xx}\;\mathrm{and}\;D_{yy}$). Increasing the magnetic force density would accelerate the MNPs and lead to more violent collisions between the MNPs and the axons. Although this increases the velocity of MNPs that successfully pass through the gaps between axons and lead to increased $D_{xx}\;\mathrm{and}\;D_{yy}$, more MNPs would be blocked due to their higher velocity and more violent collisions with the axons; these accelerated MNPs would then turn in the *Z* direction, thus dramatically increasing $D_{zz}$ and the diffusion anisotropy.4.The impact of collisions with axons can also be found for 70 nm MNPs, as the component of the diffusion coefficient in the Z direction ($D_{zz}$) significantly increases with the magnetic force density. Because the MNPs are still smaller than the average distance between axons, the rebound particles can travel in all three directions of X, Y, and Z. Since it is a random event whether the x-component overpowers the y-component, $D_{xx}\;\mathrm{and}\;D_{yy}$ alternately outpace each other, as shown in Figure 4D.

Figure 4C shows that increasing magnetic force density has a significantly limited effect on 50 nm MNPs, and the isotropic D could be achieved when χB∇B=2000. We chose $f=2000/\mu_{0}$ to draw the relationship between particle size and D, and found that the isotropic D could also be achieved when d≈80 nm under the same magnetic force density, which provides more flexibility to clinical applications.

### 3.4. Effect of Axon Shape (Tortuosity) and Magnetic Force Density

As shown in Figure 1B, the majority of axons have tortuosities ranging from 1 to 1.3. Therefore, in this study, we conducted analyses on microstructures with four different tortuosities, namely, τ=1.0,1.1,1.2,1.3, respectively. Figure 5 shows the effect of magnetic force density on the D in different microstructures. To focus on the effect of axon shape, the MNP size was fixed at 50 nm. The direction of the magnetic field was perpendicular to the Z direction.

Initially, when f=0, $D_{xx}\approx D_{yy}<D_{zz}$ in all microstructures, but the difference significantly decreases with the tortuosity; when τ=1.3, $D_{xx}\;\mathrm{and}\;D_{yy}$ are even comparable to $D_{zz}$, the approximately isotropic diffusion can be automatically achieved. The effect of τ becomes more complex after applying the external magnetic field. $D_{xx}\approx D_{yy}$ always holds for τ=1.0 regardless of the magnetic force density. In contrast, $D_{xx}$ increases faster with the magnetic force density compared to $D_{yy}$ when τ>1.0. This is because only the MNP movement in the XY direction was accelerated when τ=1.0, as shown in Figure 5E, whereas the diffusion in the Z direction was less affected (see Figure 5A). However, when τ>1.0, as axons bend towards the X direction in the present microstructures reconstructed based on Figure 1 of Ref. [25], the MNPs can more easily change direction and move along the axon tracts when encountering resistance in the XY direction. Under this condition, the acceleration direction is in the XY direction, but slightly towards the Z direction, as shown in Figure 5F. This trend can be enhanced by either increasing resistance in the XY direction (i.e., acceleration introduced by higher *f*) or increasing τ. These findings explain why $D_{zz}$ in Figure 5B–D increases with *f* and increases more sharply with τ. Furthermore, as axons bend towards the X direction in the reconstructed microstructures, the derivative Z acceleration has an X component; thus, $D_{xx}>D_{yy}$. This effect is enhanced with tortuosity, hence the difference between $D_{xx}\;\mathrm{and}\;D_{yy}$ also increases with tortuosity.

### 3.5. Effect of Magnetic Field Direction and Magnetic Force Density

As analysed above, the unexpected anisotropy in the X and Y directions is due to the change in acceleration direction, which is caused by axon bending (see Figure 5F). In the following analyses, we tried to change the direction of the magnetic field to minimise this effect. Shown in Figure 6 are the effects of magnetic field direction on Ds of 50 nm MNPs in microstructures with different tortuosities. The direction of the magnetic field is perpendicular to the Z direction in the top panels and perpendicular to the local axon tracts in the bottom panels.

As shown in Figure 6(A2–C2), applying the magnetic field perpendicular to the local axons can effectively eliminate the anisotropy of the diffusion coefficient in the X and Y directions. Regardless of the axon shape, the isotropic D≈20μm^2^/s can be achieved in all the tested microstructures using similar magnetic force densities of approximately $3000/\mu_{0}$. The comparison with Figure 6(A1–C1) denotes that tortuosity has a limited impact on how the external magnetic field manipulates the MNPs in the brain WM when the magnetic field is perpendicular to the local axons.

## 4. Discussion

In the absence of advanced imaging techniques to precisely capture the diffusion behaviours of MNPs manipulated by an external magnetic field in vivo, we adapted a mathematical model to capture the controlled diffusion behaviours of MNPs in 3D microstructures under an external magnetic field. The parametric analyses conducted in this study based on the mathematical model provide important qualitative insights into how the major parameters in the delivery system affect the diffusion behaviours of the MNPs in brain WM, which possess great clinical importance [41,42]. So far, we have shown that the diffusion coefficient of NPs in the brain WM can be significantly increased by surface charge [24], and their diffusion direction can be controlled by applying MNPs together with an external magnetic field. However, more evidence from dedicated experiments potentially enabled by high-resolution and high-frame-rate imaging techniques capable of tracking NP diffusion behaviours in the brain parenchyma is still needed to further consolidate the findings of this study.

While greater attention has been paid to improving the chemical and biological performances of NPs, e.g., reducing drug elimination rate [43] and increasing cytotoxic effects [44], to enhance the effectiveness of NPs, we found that MNP size and the external magnetic field are of vital importance to achieving the desired spatial distribution of MNPs in the brain microstructure. Instead of identifying the individual impact of each factor, we found that the particle magnetic susceptibility (χ), magnetic field intensity (*B*), and its gradient (∇B) work as a group with the factor of particle size (*d*) to influence MNP diffusion. Results from this study provide flexibility to achieve the desired diffusion phenomenon by optimising a combination of the abovementioned factors that suit the clinical practice best. For instance, although a much higher magnetic force density is required to drive smaller MNPs to reach uniform distribution (see Figure 4), the magnetic field intensity and its gradient may not necessarily need to increase because smaller MNPs could have a much higher magnetic susceptibility [45] (see Equation (Equation 3)). Moreover, the modelling results also demonstrate that if the particle size is much smaller than the gaps between axons, applying an external magnetic field perpendicular to the local axons enables isotropic diffusion. However, an isotropic D may not exist if the particle size is comparable to the gaps between axons, since the applied external magnetic field would drastically increase $D_{zz}$ over $D_{xx}\;\mathrm{and}\;D_{yy}$. Although magnetic tools may not be able to help large MNPs achieve uniform distribution in WM, it is still feasible to increase their transverse penetration as $D_{xx}\;\mathrm{and}\;D_{yy}$ increase with the magnetic force density. One needs to note that the tissue microstructure can vary considerably between individuals depending on the location of the lesion in the brain, the patient’s age, gender, and disease stage, etc. Such complexity would require a personalized treatment plan to maximize the delivery outcomes.

Figure 5 and Figure 6 show that although axon tortuosity together with magnetic force density enhances the transverse diffusion anisotropy of MNPs in WM, placing the magnetic field perpendicular to the local axons can make the diffusion coefficient isotropic. For the situation shown in Figure 1 of Ref. [25], more attention should be paid to adjusting the direction of the applied magnetic field. However, in some other regions, where axon bending direction is more random, transverse anisotropy introduced by axon tortuosity may not be as significant as it appears in this study. With the aid of the diffusion tensor imaging (DTI) technique [46], the principal direction of axon tracts can be obtained. This can be used as a key reference to determine the direction when applying an external magnetic field.

As mentioned at the beginning of this article, the nature of drug molecules and the blood–brain barrier makes it difficult for drugs to cross the blood–brain barrier. This mathematical framework is actually a generic framework that can visualise the particle transport process in the biological environment by mathematically describing the particle–particle interaction, particle–fluid interaction, particle–interface interaction, and the particles’ thermal motions. Therefore, it has the potential to also model the process of drug molecules passing through the BBB and provide useful suggestions to design novel drugs. This could be achieved by (i) adding blood vessels to the geometry, and (ii) developing the governing equations to describe this transvascular transport for both fluid and particle. Furthermore, the present mathematical model (governing equations) possesses the potential to consider other properties of nanoparticles, e.g., shape, size, surface charge, and composition, as the theoretical principle is to describe the forces that control the particles’ trajectories by mathematical formulas, while the governing equations in the present mathematical model include some of the most general forces acting on nanoparticles. However, to explicitly consider particle shape, the geometric model of the particles needs to be modified. To consider the particle’s surface characteristics, e.g., surface charge, more governing equations that are able to describe the electric forces need to be added to this mathematical model and solved together. In our previous study [24], the effect of surface charge on the diffusion behaviours of NPs was investigated. These are the same as the composition and porosity of the nanoparticle. The impacts of these variables will be studied in future.

Finally, some assumptions and limitations in the present study deserve further discussion. Regarding the mathematical and geometrical models, several factors that can also affect the transport of MNPs in the brain WM are not taken into account; these include the water transport across the cell membrane, hydrophobic nature of large biomolecules, the variation in local fluid viscosity due to the components in the extracellular matrix, and MNP–cell adhesion. This is mainly because there is a lack of mathematical models that can accurately describe and reconcile these complex processes. Further support from experiments is also needed to establish appropriate models for these processes in future. In this model, the particles were treated as spheres, while NPs in other shapes, e.g., nanotube, nanodisk, nanoneedle, plateloid, and ellipsoid, are also widely adopted as drug carriers [47]. However, the proposed mathematical models could be readily adapted to investigate the effect of MNP shape on their diffusion behaviour in the brain by constructing NPs with different shapes. The WM cross-section is swept along the axon tract to generate the 3D microstructure, as shown in Figure 2. The realistic 3D structure could be more complex, and the directions of the axons may not be so uniform, depending on the location in the brain. However, as the aim of this study was to qualitatively understand the effects of parameters of the magnetic system on the diffusion tensor of MNPs, the representative microstructure is enough to provide key findings. The 3D realistic microstructures rebuilt from microscopic images can be used in future studies on specific degenerative nerve diseases. Furthermore, this mathematical modelling framework has the potential to numerically characterise the equivalent diffusion coefficients of different types of NPs in the brain white matter; nonetheless, dedicated experiments are needed to validate the precision. By continuously improving this mathematical framework, we anticipate that we will be able to precisely tune the diffusion coefficient and diffusion direction of NPs, which would remarkably increase the efficiency of delivering nanodrugs into the deep brain tissues.

## 5. Conclusions

In summary, this modelling framework enables us to gain a deeper understanding on (i) 3D diffusion behaviours of MNPs in brain WM and the corresponding diffusion coefficient tensors, (ii) how MNP diffusion behaviours in the anisotropic tissue can be manipulated by the externally applied magnetic field, and (iii) how particle size, particle susceptibility, magnitude and direction of the externally applied magnetic field, and axon shape affect the MNPs’ equivalent diffusion coefficient tensors. The following key findings were obtained from the present study:1.Applying an external magnetic field could achieve uniform distribution (isotropic D) of MNPs in anisotropic tissues when the particle size is much smaller than the gaps between cells. We thus anticipate that applying a complex magnetic field may potentially determine the spatial distribution pattern.2.When the particle size is comparable to the gaps between cells, isotropic D of MNPs cannot be achieved. The anisotropy even increases with the external magnetic field.3.Special attention should be paid to the particle size, as the selection of particle size would affect the settings of nearly all the other key parameters in the whole system.4.The magnetic field should be perpendicular to the local axon tracts to eliminate the transverse anisotropy of D induced by axon tortuosity.5.The parameter χB∇B (equal to $f/\mu_{0}$) could work as a derived parameter to influence the MNPs’ equivalent diffusion coefficient tensor. Adopting this derived parameter would provide more flexibility to the practical applications.

## Figures and Tables

**Figure 1 ijms-24-02534-f001:**
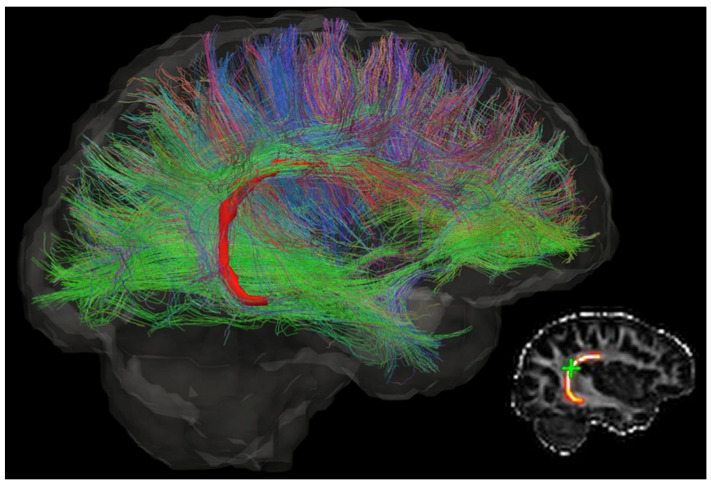
The diffusion tensor image of a brain, which shows the direction of neurons and complexity of the brain microstructure. Different colours indicate different directions. This figure is reprinted from Ref. [12] with open access under the terms of the CC BY-NC-ND 4.0 License.

**Figure 2 ijms-24-02534-f002:**
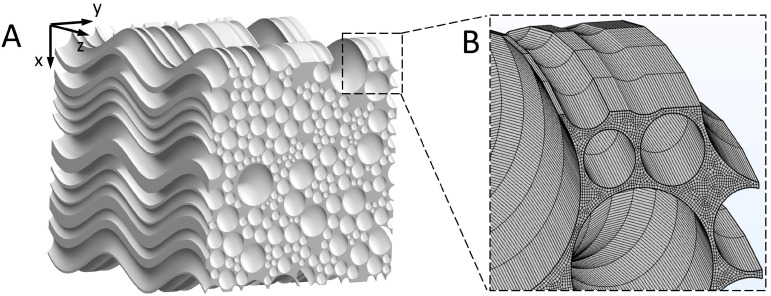
3D reconstruction of the WM microstructure. (**A**) Reconstructed microstructure of the brain WM. (**B**) Finite element mesh of the microstructure.

**Figure 3 ijms-24-02534-f003:**
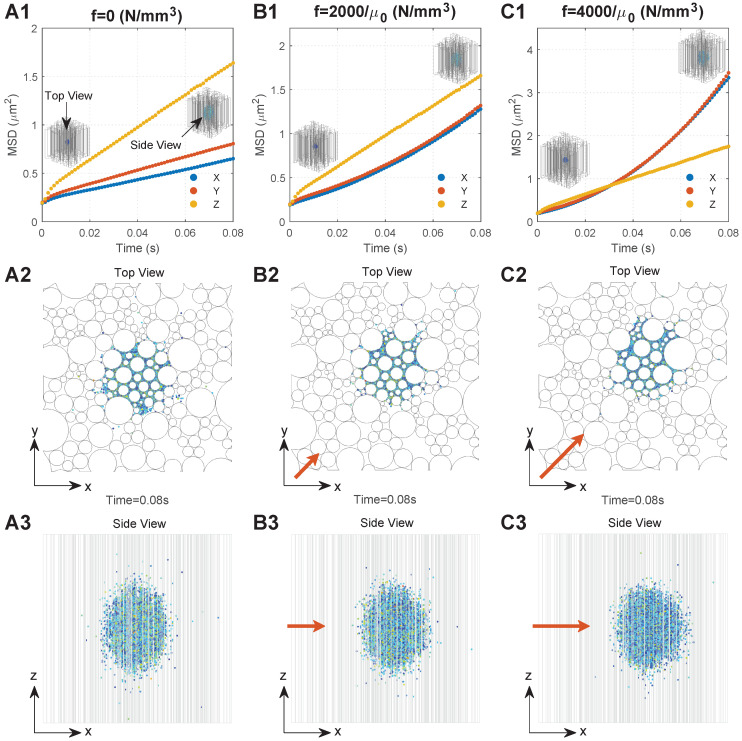
Typical diffusion behaviours of the MNPs in the brain WM under different conditions. Series A. Results when there is no external magnetic field: (**A1**) relationship between MSD and time; (**A2**) top view of MNP distribution; (**A3**) side view of the MNP distribution. Series B. Results when the magnetic force density is $2000/\mu_{0}$: (**B1**) relationship between MSD and time; (**B2**) top view of MNP distribution; (**B3**) side view of the MNP distribution. Series C. Results when the magnetic force density is $4000/\mu_{0}$: (**C1**) relationship between MSD and time; (**C2**) top view of MNP distribution; (**C3**) side view of the MNP distribution. The directions of the top view and side view are shown in (**A1**), and the direction of the applied magnetic field is shown by the orange arrows, the lengths of which are proportional to the magnetic field intensity.

**Figure 4 ijms-24-02534-f004:**
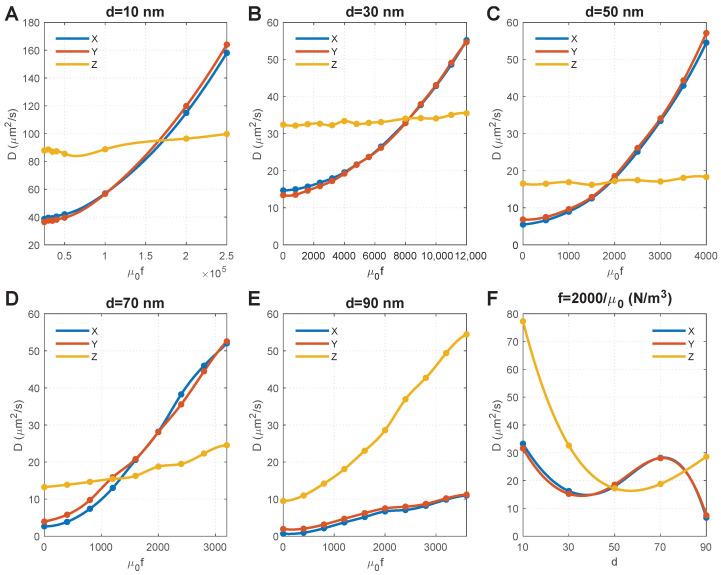
Effect of particle size and magnetic force density on the diffusion coefficients of MNPs. (**A**–**E**) show the results of MNPs with diameters of 10 nm, 30 nm, 50 nm, 70 nm, 90 nm, respectively. Note that MNPs with different particle sizes have different kinetic energies under the same magnetic force density; thus, different magnetic force densities were applied to MNPs with different particle sizes in order to capture the points where diffusion coefficients in the X and Y directions are equal to that in the Z direction. (**F**) Relationship between diffusion coefficients and particle size when the magnetic force density is $2000/\mu_{0}$.

**Figure 5 ijms-24-02534-f005:**
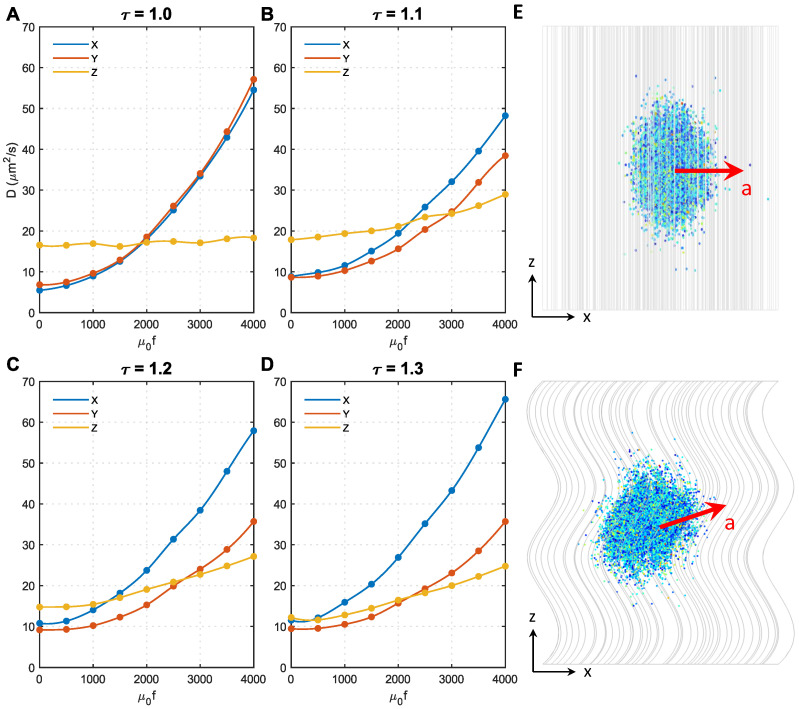
Effect of axon shape (tortuosity) and magnetic force density on the diffusion coefficients of MNPs. (**A**–**D**) show the results of axon shape with the tortuosity of 1.0, 1.1, 1.2, 1.3, respectively. (**E**) The acceleration direction of 50 nm MNPs in microstructure with τ=1.0. (**F**) The acceleration direction of 50 nm MNPs in microstructures with τ>1.0.

**Figure 6 ijms-24-02534-f006:**
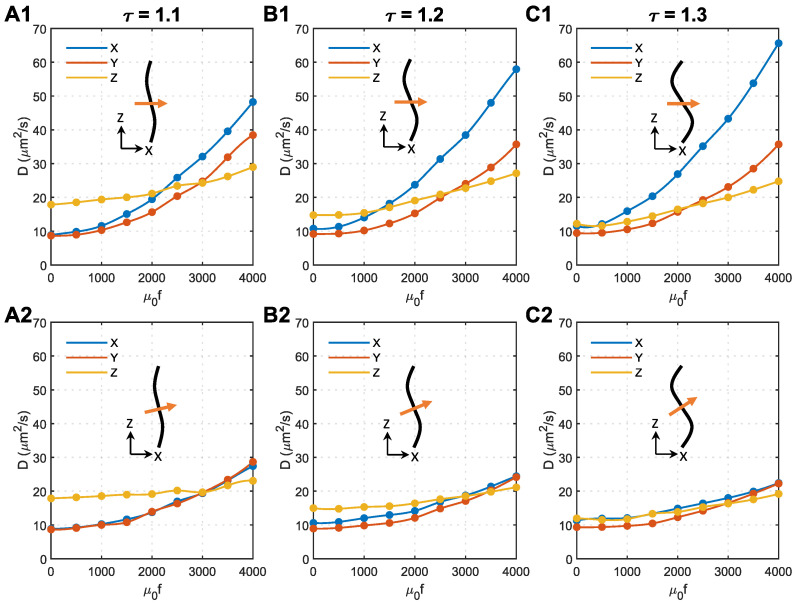
Effect of magnetic field direction on the anisotropy of MNP diffusion coefficients in the brain WM. (**A1**) Tortuosity = 1.1, the magnetic field is perpendicular to the Z direction. (**A2**) Tortuosity = 1.1, the magnetic field is perpendicular to the axons. (**B1**) Tortuosity = 1.2, the magnetic field is perpendicular to the Z direction. (**B2**) Tortuosity = 1.2, the magnetic field is perpendicular to the axons. (**C1**) Tortuosity = 1.3, the magnetic field is perpendicular to the Z direction. (**C2**) Tortuosity = 1.3, the magnetic field is perpendicular to the axons. In each subfigure, the black curve represents the axon shape, while the orange arrow shows the direction of the applied magnetic field.

**Table 1 ijms-24-02534-t001:** Comparison between experimental measurements and simulation results.

Tissue	$d_{H}$ (nm)	χ	B (T)	Magnet Length (cm)	$D_{\mathrm{Exp}}$ ($10^{-11}$ m${}^{2}$/s)	$D_{\mathrm{Sim}}$ ($10^{-11}$ m^2^/s)
Blood [34,35]	10	4.66	1.3	6	4	4.49
Blood [35,36]	100	6.11	0.8	3	60	62.6
Brain WM [37]	2	-	-	-	20	22.2

## Data Availability

Data available on request from the authors.

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
