# Peer review of "Mathematical Optimisation of Magnetic Nanoparticle Diffusion in the Brain White Matter"

_ijms, 2023, doi:10.3390/ijms24032534_

Round 1

Reviewer 1 Report

ABSTRACT

Abstract is very unspecific, and it are showing only general information of results. Some representative specific quantitative data should be showed.

More that 30% is introduction. It should be written in a more direct way with shorter introduction and going more directly to show what has been done and more specific data of results.

GENERAL CONSIDERATION OF THE PAPER

Until I understand, this is a full theoretical approach. I have no objection in the strategy for modelling.

However, if the in-silico modelling is not contracted with a minimum of actual experimental data, I cannot appreciate the validation of the approach.

 Therefore, no further evaluation of the manuscript can be performed.

I suggest being re-written the manuscript, introducing some actual experimental data before considered for publication

Author Response

We would like to thank the reviewer very much for providing these useful suggestions. Below we have addressed the reviewer’s concerns point-by-point and modified the manuscript accordingly.

Point 1: Abstract is very unspecific, and it are showing only general information of results. Some representative specific quantitative data should be showed.

Response 1: We have revised the abstract by providing more quantitative data and stating more specific results that we have obtained in the present study. The modified abstract is:

Magnetic nanoparticle (MNPs) is a promising drug delivery system to treat brain diseases as the particle transport trajectory can be manipulated by an external magnetic field. However, due to the complex microstructure of brain tissues, particularly the arrangement of nerve fibres in the white matter (WM), how to achieve desired drug distribution patterns, e.g. uniform distribution, is largely unknown. In this study, by adopting a mathematical model capable of capturing the diffusion trajectories of MNPs, we conducted a pilot study to investigate the effects of key parameters in the MNP delivery on the particle diffusion behaviours in the brain WM microstructures. The results show that (i) a uniform distribution of MNPs can be achieved in anisotropic tissues by adjusting the particle size and magnetic field; (ii) particle size plays a key role in determining MNPs’ diffusion behaviours. The magnitude of MNP equivalent diffusivity is reversely correlated to the particle size. The MNPs with a dimension greater than 90 nm cannot reach a uniform distribution in the brain WM even in an external magnitude field; (iii) axon tortuosity may lead to transversely anisotropic MNPs' transport in the brain WM; but this effect can be mitigated by applying an external magnetic field perpendicular to the local axon track. This study not only advances understanding to answer the question of how to optimise the MNP delivery but also demonstrates the potential of mathematical modelling to help achieve desired drug distributions in biological tissues with a complex microstructure.

Point 2: Abstract - More than 30% is introduction. It should be written in a more direct way with shorter introduction and going more directly to show what has been done and more specific data of results.

Response 2: We have shortened the introduction and background parts in the abstract and made it more direct to show the work and the results we have obtained.

Point 3: Until I understand, this is a full theoretical approach. I have no objection in the strategy for modelling. However, if the in-silico modelling is not contracted with a minimum of actual experimental data, I cannot appreciate the validation of the approach. Therefore, no further evaluation of the manuscript can be performed. I suggest being re-written the manuscript, introducing some actual experimental data before considered for publication.

Response 3: We are sorry for not explicitly providing validation of the modelling method in the presence of magnetic field.  The proposed method has already been validated with experimental findings in the absence of magnetic field in our previous contributions. This work is a pilot study to demonstrate the concept of optimising the delivery efficiency of magnetic nanoparticles (MNPs) in the brain parenchyma after passing through the BBB by means of mathematical modelling in the absence of advanced imaging techniques capable of precisely capturing the diffusion behaviours of MNPs manipulated by an external magnetic field in vivo, as stated in the last paragraph of Introduction and 1st paragraph of Discussion. Consequently, there is a lack of dedicated experiments in the open literature to compare against the findings in this study at this stage, and the experimental investigation is also outside the scope of the present study.

However, we agree that experimental validation is necessary and would like to thank the reviewer for this constructive suggestion. In the revised manuscript, we have added Section 3.1: Model validation, where we have used our theoretical model to simulate MNPs diffusion experiment in the blood and a non-magnetic NPs diffusion experiment in the brain white matter. We believe that comparison against the MNPs diffusion coefficient in the blood helps demonstrating the validity of the mathematical mode, and the NPs diffusion coefficient in the brain white matter corroborates the precision of the mathematical model in the absence of the magnetic force term and geometric model in the present study. As the theoretical diffusion coefficients agree well with the measured diffusion coefficients, the accuracy of the present theoretical model have been validated in the revised manuscript.

To avoid further misunderstandings and make the theoretical model more rigorous, we have modified the manuscript to add model validation, specify the scope of the present study, and highlighted the limitations, as listed below:

  1. Modified the Abstract:

Here, in this study, by adopting a mathematical model capable of capturing the diffusion trajectories of MNPs in the brain microstructures, we conducted a pilot study to investigate the effects of key parameters in the MNPs delivery system on the diffusion behaviours of MNPs in the microenvironment of brain WM.

  1. Added the Section 3.1:

Due to the absence of experimental measurement on the diffusion coefficient of MNPs in the brain microenvironment, we validated the present theoretical model by using it to model the experiments reported in Ref. [34 – 37]. In Ref. [34– 36 ], the diffusion behaviours of 10 nm and 100 nm MNPs in blood were monitored and the corresponding Ds of the MNPs were calculated. Comparison against these data will validate the precision of the mathematical model. In Ref. [37], the D of gadobutrol (cerebrospinal fluid (CSF) tracer with the hydraulic diameter (dH) of 2 nm) in the brain WM were calculated based on MRI analysis and partial differential constrained optimisation. Although non-magnetic NPs were applied, this experiment is able to validate the precision of the mathematical model excluding the magnetic force term and geometric model in the present study. Therefore, we chose these experimental data to conduct the model validation study. The experimental parameters and their respective Ds are summarised in Table 1.

By calculating the trajectories of the individual NPs using the mathematical model, we obtained the time-MSD curves of the NPs in each experiment (see Fig. 3). According to Eq. 5, the slopes of the curves’ stable phase were used to calculate the theoretical equivalent diffusion coefficient [ 24 ], as presented by  in Table 1. The comparisons in Table 1 show that the theoretical results agree well with the experimental data. It is worth mentioning that the Ds in experiments are smaller than the Ds in the simulations. This should be attributed to the simplification of the mathematical model as the resistance to particle movement from other factors, e.g. proteins and fibres, in the real micro-environment cannot be explicitly described. However, this limitation can be partly mitigated by adopting the viscosity of the measured interstitial fluid, as mentioned in Section 2.3.

  1. Added some discussions, 1st paragraph in Discussion:

However, more evidence from dedicated experiments potentially enabled by high-resolution and high frame-rate imaging techniques capable of tracking NPs' diffusion behaviour is still needed to further consolidate the findings in this study.

  1. Last paragraph in Discussion:

Furthermore, this mathematical modelling framework has the potential to numerically characterise the equivalent diffusion coefficient of different types of NPs in the brain white matter, but more dedicated experiments are needed to validate the precision.

Reviewer 2 Report

The present study adopted a mathematical model capable of capturing the diffusion trajectories of magnetic nanoparticles (MNPs) in the cerebral microenvironment. Key parameters in the delivery system of MNPs in the equivalent diffusion coefficient of MNPs in the brain WM microenvironment were investigated.

The manuscript is well written, clearly presented and scientifically acceptable. I have some comments:

1) Depending on the nature of the drug, its greatest difficulty is to cross the blood-brain barrier. Can the mathematical model built be useful, to some extent, to solve this problem? Please include any comments about this in the discussion.

2) The shape, size, characteristics (such as porosity) of the surface, surface charge and composition of nanoparticles can directly influence the diffusion of nanomaterials in biological microenvironments. Would the proposed mathematical model be applicable if we considered these variables?

3) Please include in the manuscript the limitations that the authors are able to identify for the proposed model.

Author Response

The present study adopted a mathematical model capable of capturing the diffusion trajectories of magnetic nanoparticles (MNPs) in the cerebral microenvironment. Key parameters in the delivery system of MNPs in the equivalent diffusion coefficient of MNPs in the brain WM microenvironment were investigated. The manuscript is well written, clearly presented and scientifically acceptable. I have some comments:

We would like to thank the reviewer very much for appreciating the merits of the present study and providing the very useful suggestions. Below we have addressed the reviewer’s concerns point-by-point and modified the manuscript accordingly.

Point 1: Depending on the nature of the drug, its greatest difficulty is to cross the blood-brain barrier. Can the mathematical model built be useful, to some extent, to solve this problem? Please include any comments about this in the discussion.

Response 1: This mathematical model is a generic framework that can visualise the particle transport process in the biological tissue microstructure by mathematically describing the particle-particle interaction, particle-fluid interaction, particle-interface interaction, and particles’ thermal motion. Therefore, the present mathematical model indeed has the potential to also model the process of drug molecules passing through the BBB and provide useful suggestions to design novel drugs. The modification of the current model would then be changing the model geometry and, potentially, some of the acting forces dependent on the specific microenvironment. We very much appreciate this comment and have added some discussions as shown below:

As mentioned at the beginning of this article, the nature of drug molecules and the blood-brain barrier makes it difficult for drugs to enter the brain to cross the blood-brain barrier. This mathematical framework is actually a generic framework that can visualise the particle transport process in the biological environment by mathematically describing the particle-particle interaction, particle-fluid interaction, particle-interface interaction, and particles’ thermal motion. Therefore, it has the potential to also model the process of drug molecules passing through the BBB and provide useful suggestions to design novel drugs. This could be achieved by (i) adding blood vessel to the geometry, and (ii) developing the governing equations to describe this transvascular transport for both fluid and particle.

Point 2: The shape, size, characteristics (such as porosity) of the surface, surface charge and composition of nanoparticles can directly influence the diffusion of nanomaterials in biological microenvironments. Would the proposed mathematical model be applicable if we considered these variables?

Response 2: Yes, this mathematical model (to be more specifically, the governing equations) possesses the potential to consider other properties of nanoparticles, since the principle is to describe the forces that control the particles trajectories by mathematical formulas while the governing equations in the present mathematical model have included some of the most general forces acting on nanoparticles. However, to explicitly consider particle shape, the geometric model of the particles need to be modified. To consider the particle’s surface characteristics e.g. surface charge, some more governing equations which are able to describe the electric forces need to be added to this mathematical model and solved together. In our previous study [24], the effect of surface charge on the diffusion behaviours of NPs has been investigated. These are the same as the composition and porosity of the nanoparticle. The impacts of these variables will be studied in future. We very much appreciate this comment and have added some discussions as shown below:

Furthermore, the present mathematical model (governing equations) possesses the potential to consider other properties of nanoparticles e.g. shape, size, surface charge and composition, as the theoretical principle is to describe the forces that control the particles' trajectories by mathematical formulas while the governing equations in the present mathematical model have included some of the most general forces acting on nanoparticles. However, to explicitly consider particle shape, the geometric model of the particles needs to be modified. To consider the particle’s surface characteristics e.g. surface charge, some more governing equations which are able to describe the electric forces need to be added to this mathematical model and solved together. In our previous study [24], the effect of surface charge on the diffusion behaviours of NPs has been investigated. These are the same as the composition and porosity of the nanoparticle. The impacts of these variables will be studied in future.

Point 3: Please include in the manuscript the limitations that the authors are able to identify for the proposed model.

Response 3: We very much appreciate this comment. Although we have discussed some of the limitations in the original form of the manuscript, such as the simplification of the effects from the extracellular matrix components and the idealisation of the brain microstructure, below we have added some more discussions about the limitations and prospective solutions inspired by the reviewer in the last paragraph of the revised manuscript:

  1. In this model, the particles were treated as spheres, while NPs in other shapes e.g. nanotube, nanodisk, naoneedle, plateloid, and ellipsoid are also widely adopted as drug carriers [43]. However, the proposed mathematical models could be readily adapted to investigate the effect of MNPs' shape on their diffusion behaviours in the brain by constructing NPs with different shapes.
  2. Furthermore, this mathematical modelling framework has the potential to numerically characterise the equivalent diffusion coefficient of different types of NPs in the brain white matter, but advanced experiments are needed to validate the precision.

Round 2

Reviewer 1 Report

I appreciate the changes introduced and how the authors have taken into consideration my comments and suggestions.